# Oxidative Properties of Polystyrene Nanoparticles with Different Diameters in Human Peripheral Blood Mononuclear Cells (In Vitro Study)

**DOI:** 10.3390/ijms22094406

**Published:** 2021-04-23

**Authors:** Kinga Kik, Bożena Bukowska, Anita Krokosz, Paulina Sicińska

**Affiliations:** Department of Biophysics of Environmental Pollution, Faculty of Biology and Environmental Protection, University of Lodz, Pomorska Str. 141/143, 90-236 Lodz, Poland; kinga.kik@edu.uni.lodz.pl (K.K.); bozena.bukowska@biol.uni.lodz.pl (B.B.); anita.krokosz@biol.uni.lodz.pl (A.K.)

**Keywords:** polystyrene nanoparticles, ROS, lipid and protein oxidation, apoptosis, cells viability, LC_50_

## Abstract

With the ongoing commercialization, human exposure to plastic nanoparticles will dramatically increase, and evaluation of their potential toxicity is essential. There is an ongoing discussion on the human health effects induced by plastic particles. For this reason, in our work, we assessed the effect of polystyrene nanoparticles (PS-NPs) of various diameters (29, 44 and 72 nm) on selected parameters of oxidative stress and the viability of human peripheral blood mononuclear cells (PBMCs) in the in vitro system. Cells were incubated with PS-NPs for 24 h in the concentration range of 0.001 to 100 µg/mL and then labeled: formation of reactive oxygen species (ROS) (including hydroxyl radical), protein and lipid oxidation and cell viability. We showed that PS-NPs disturbed the redox balance in PBMCs. They increased ROS levels and induced lipid and protein oxidation, and, finally, the tested nanoparticles induced a decrease in PBMCs viability. The earliest changes in the PBMCs were observed in cells incubated with the smallest PS-NPs, at a concentration of 0.01 μg/mL. A comparison of the action of the studied nanoparticles showed that PS-NPs (29 nm) exhibited a stronger oxidative potential in PBMCs. We concluded that the toxicity and oxidative properties of the PS-NPs examined depended to significant degree on their diameter.

## 1. Introduction

Plastics are now a serious environmental problem in the world. Since the massive production of plastics started in the 1950s, the output has been rising annually. In the 1950s, approximately 1.5 million tons of plastics were produced; the amount has been raised 200-fold up to 299 million tons worldwide in 2013, and from the years 2015 to 2016 from 322 to 335 million tons a year [1]. It has been estimated that the amount of produced plastics may increase significantly and even be doubled in the coming years [2]. Approximately 90% of the total amount of plastics consists of high-density polyethylene, low-density polyethylene, polyvinyl chloride, polystyrene, polypropylene and polyethylene terephthalate [3]. Numerous studies have shown that the degradation of plastic leads to the release of nanoparticles (NPs) into the environment [4,5]. Plastic may be degraded into microparticles (MPs) < 5000 nm (5 μm) in diameter, and further into NPs < 100 nm (0.1 μm) in diameter [6]. Plastic NPs may enter human organisms in food, water, air and through skin. Then, they accumulate in subsequent links of the food chain [7,8,9].

One of the main plastics used for packaging, commercial and construction purposes is polystyrene (PS) [10]. It is a petroleum-based material obtained by the polymerization of styrene (vinylbenzene) monomers. According to the US Environmental Protection Agency (EPA), the polystyrene manufacturing process is the fifth largest source of hazardous waste. Despite the existing evidence regarding the toxicity of styrene, the styrene polymerization product—polystyrene—is not listed as hazardous in any policy documents.

Plastic nanoparticles, including polystyrene, enter the human body mainly through the digestive and respiratory tracts [11,12]. Depending on the cell type and NP size, they may be transported through phagocytosis, pinocytosis, macro-pinocytosis or absorbed by passive transport, and as a consequence they can enter cellular membranes and various biological structures [13]. In rats, after oral administration, 6% of PS particles (0.87 μm) were detected in the circulation within 15 min [11], whereas oral exposure to polystyrene nanoparticles (1.25 mg kg^−1^, size 50 nm) resulted in 34% absorption, most probably transported through the mesentery lymph to reach the circulatory system and accumulate mostly in the liver [12]. Moreover, translocation across the mammalian gut into the lymphatic system of various types and sizes of microparticles (between 0.1 and 150 mm) has been demonstrated in studies involving humans (0.2 and 150 mm) [14]. It is known that particles <110 μm can enter the blood stream through the portal vein and particles <20 μm can reach internal organs [15,16,17]. Crossing the intestinal barrier, particles <100 nm can even be transported into the brain and across the placental barrier [17,18,19,20,21,22]. Recently, Ragusa et al. (2021) showed in their research the presence of 12 microplastic fragments (ranging from 5 to 10 μm in size), with spheric or irregular shape in four placentas from six placentas investigated in humans [23]. As shown in the above research, inhaled particles can be excreted by mucociliary clearance, but can also settle in the lungs or be absorbed into the bloodstream [17,24]. PS-NPs can enter the circulatory system and thus may interact with all blood components.

There is little data on the effects of polystyrene nanoparticles on blood cells. Studies conducted on workers exposed to styrene showed a decreased percentage of Ti helper lymphocytes and an increased ratio of suppressor T lymphocytes. These studies demonstrated the immunotoxicity of styrene [25]. Other studies suggest that exposure to styrene may cause lymphoma and leukemia in humans, while polystyrene has not been shown to be carcinogenic [26]. Currently, most studies on the toxicity of plastic particles are animal studies. It was observed that polystyrene induces oxidative stress in *Danio rerio* [27] and mice [28,29,30] and increases the ROS level in *Daphnia magna* [31].

Insufficient data on the harmful effects of polystyrene nanoparticles prompted us to undertake studies using alternative in vitro tests to elucidate the cellular mechanisms of toxicity. To increase the relevance of these tests for humans, the European Union Reference Laboratory for Alternatives to Animal Testing (EURL-ECVAM) recommends the use of human cells for all in vitro test systems. The source of the cells may be peripheral human blood. It is the first target model for exposure to chemicals of environmental or industrial origin and it provides information on the body’s overall response to xenobiotics [32,33,34].

In our study, we used peripheral human blood mononuclear cells, which were incubated with polystyrene nanoparticles of various sizes (29, 44 and 77 nm) for 24 h in the concentration range of 0.001 to 1000 µg/mL. We then assessed the increase in reactive oxygen species levels (including hydroxyl radical), the degree of lipid and protein oxidation and the percentage of viable PBMCs.

## 2. Results

### 2.1. Characterization of Polystyrene Nanoparticles

The PS-NPs size distribution in phosphate-buffered saline (pH 7.4) is presented in Figure 1 and Table 1. As we can see, the diameters of the particles indicated by the manufacturer are consistent with the results measured by Dynamic Light Scattering (DLS) method.

### 2.2. Reactive Oxygen Species

24-h incubation of PBMCs with polystyrene nanoparticles showed an increase in ROS in the cells tested. All tested particles caused oxidation of the fluorescent 2′-7′-dichlorodihydrofluorescein probe. Statistically significant changes versus control were caused by PS-NPs with a diameter of 29 and 44 nm, as they increased the production of reactive oxygen species at a concentration of 0.01 µg/mL. On the other hand, nanoparticles with a diameter of 72 nm caused statistically significant changes versus control from at the concentration of 0.1 µg/mL (Figure 2).

### 2.3. Hydroxyl Radical Level

The oxidation of 4-hydroxy-phenyl- fluorescein (HPF) is associated with the formation of highly reactive oxygen species including the hydroxyl radical. Statistically significant changes in HPF oxidation were observed after the 24-h incubation of PBMCs with all analyzed nanoparticles, 29, 44 and 72 nm in diameter. It was shown that the largest changes were induced by PS-NPs with the smallest diameter from 1 µg/mL. The 44 and 72 nm nanoparticles caused a statistically significant increase in the level of the hydroxyl radical at the concentration of 10 µg/mL (Figure 3).

### 2.4. Lipid Peroxidation

Assessment of lipid peroxidation in PBMCs treated with polystyrene nanoparticles of various diameters in the concentration range of 0.001 to 100 µg/mL was performed. All analyzed nanoparticles caused a significant decrease in the fluorescence of cis-parinaric acid. It was shown that the largest changes were shown by nanoparticles with a diameter of 29 nm, which caused a statistically significant decrease in the examined parameters versus the control at the concentration of 0.1 µg/mL. However, in the case of larger polystyrene particles with a diameter of 44 and 72 nm, statistically significant changes versus control were observed at the concentration of 1 µg/mL (Table 2).

### 2.5. Protein Damages

All tested nanoparticles were reported to cause oxidative damage to proteins in lymphocytes after 24 h of incubation. The 29 and 44 nm nanoparticles induced a statistically significant decrease in tryptophan fluorescence at the concentrations of 0.1 and 1 µg/mL, respectively, compared to the control sample. In the case of the largest nanoparticles, the changes were statistically significant from 10 µg/mL (Table 2).

### 2.6. Cell Apoptosis

All PS-NPs increased the number of apoptotic cells. Nanoparticles of 29 nm in diameter starting from the concentration of 1 µg/mL caused a statistically significant increase in the number of apoptotic cells. However, nanoparticles with a size of 44 and 72 nm starting from the concentrations of 10 and 100 µg/mL, respectively, caused an increase in the number of apoptotic PBMCs (Figure 4).

### 2.7. Cell Viability

All analyzed PS-NPs decreased PBMC viability. The greatest changes of the examined parameters were observed in cells incubated with nanoparticles with a diameter of 29 nm. Statistically significant changes for nanoparticles with diameters of 29 and 44 nm were at the concentration of 500 µg/mL. The largest nanoparticles with a diameter of 72 nm caused a small but statistically significant decrease in the number of cells only at the concentration of 1000 µg/mL (Figure 5).

## 3. Discussion

Nano- and microplastics, once regarded as inert particles without toxicity, are now seen as potentially harmful to organisms depending on exposure conditions and organism susceptibility [35,36,37]. The high surface area of plastic particles may be responsible for oxidative stress, cytotoxicity and particle translocation to other tissues, while their persistent nature limits their removal from the organism, leading to chronic inflammation [38]. As shown in the literature, nano- and microplastic particles can induce the production of ROS. The formation of free radicals on NP/MP surfaces occurs by photooxidation or UV radiation through cross-linking reactions [39]. Free radicals that have been generated along the polymer chain can then bind to atmospheric oxygen and lead to the formation of polymeric peroxy radicals, while continuing to form secondary polymeric alkyl radicals [3]. Currently, the studies on oxidative stress induced by PS-NPs have been conducted mainly on animals [40,41,42,43], while there have been only a few in vitro studies performed [44,45,46].

Our research showed that PS-NPs after 24 h of incubation with PBMCs in the concentration range of 0.001 to 100 µg/mL caused an increase in ROS at a concentration of 0.01 µg/mL for PS-NPs with a size of 29 and 44 nm, while for nanoparticles with a size of 72 nm a comparable increase in ROS was observed from 0.1 µg/mL (Figure 2). In the case of the hydroxyl radical, a statistically significant increase was observed from the concentration of 1 µg/mL for the smallest polystyrene nanoparticles, while for the remaining nanoparticles, a statistically significant increase was observed only at the concentration of 10 µg/mL (Figure 3). Similar increases in ROS under the influence of PS-NPs were observed by Schirinzi et al., 2017, measuring ROS formation in cells in cerebral (T98G) and epithelial (HeLa) human cells after exposure to PS-MPs (10 mm) and PS-NPs (40 and 250 nm) at 10 mg/L [46]. Positive effects were reported by Poma et al. (2019) in the human fibroblast Hs27 cell line by using a total ROS assay kit. The increase in ROS was observed after short exposure times (less than 30 min) [44]. In contrast, other researchers did not notice ROS formation in human intestinal Caco-2 cells after exposure to PS-NPs for 24 h [45]. These negative findings are in agreement with the results reported by Rubio et al. (2020) for THP-1 cells, where no effects in ROS production were noted. Mild positive effects were observed in Raji-B cells incubated for 3 h with the highest concentration of PS-NPs of 50 μg/mL. In another study, PS-NPs were able to induce a significant increase in intracellular ROS levels in TK6 cells. The authors suggested that the effects induced by PS-NPs can be strongly dependent on the type of the cells used [47]. PS-NPs caused also an increase in ROS and changes in the activity of antioxidant enzymes in organisms such as *Daphnia pulex* [40,41], marine bacteria (*Halomonas alkaliphila*) [42] and *Danio rerio* [43]. Liu et al. (2020; 2021) conducted a study on *Daphnia pulex*. The organism was exposed to 75 nm in diameter polystyrene nanoparticles at concentrations of 0.1, 0.5, 1 and 2 mg/L. It was observed that the analyzed nanoparticles induced a ROS level increase at concentrations of 0.5, 1 and 2 mg/L, and increased the content of hydrogen peroxide (H_2_O_2_) and activated the MAPK-HIF-1/NFkB pathway in the study organisms. Increased expression of the MAPK pathway genes was demonstrated at the lowest concentrations of nanoplastics. Moreover, a decrease in the activity of antioxidant enzymes, i.e., catalase (CAT) and total and copper-zinc superoxide dismutase (SOD, CuZnSOD) was found [40,41]. The formation of oxidative stress in sea bacteria (*Halomonas alkaliphila*) exposed to polystyrene nanoparticles and amine-modified nanoparticles of 55 and 50 nm in diameter, respectively, and an initial concentration of 10 g/L was also demonstrated. Amine-modified plastic particles caused higher oxidative stress than unmodified particles [42]. The latest research showed that exposure of adult zebrafish (*Danio renio*) to 70 nm NP-PS at doses of 0.5 and 1.5 ppm for 7 days resulted in their accumulation in tissues, among others in the liver, intestines, brain and gonads, lead to disorders of lipid and energy metabolism and induced oxidative stress [43]. In other studies, an increase in the activity of CAT, SOD and GSH enzymes in visceral fat, gills and mantle was observed in *Corbicula fluminea* mussels exposed to 80 nm polystyrene nanoparticles at concentrations of 0.1, 1 and 5 mg/L [48].

The toxicity of nanoplastic particles may be dependent on their ability to translocate across the gut, enter the systemic circulation, penetrate cells and interact with biological macromolecules, such as lipids and proteins [21].

As a result of the fragmentation of microparticles into nanoparticles, the surface-to-weight ratio increases, which enables their penetration directly through lipid membranes [3]. In addition, recent studies have shown that nano- and microparticles of plastics increase cellular oxidative stress, which in turn may lead to the lipid peroxidation (LPO) of cell membranes [19,49], and consequently to the loss of plasma membrane integrity and intracellular membranes [50]. Therefore, in the next step, we checked whether polystyrene nanoparticles of different sizes affected the lipid oxidation in PBMCs.

Our research showed a statistically significant increase in lipid peroxidation at a concentration of 0.1 µg/mL in the case of polystyrene nanoparticles with a size of 29 nm, while nanoparticles with a size of 44 and 72 nm caused similar statistically significant changes at the concentration of 1 µg/mL (Table 2). Furthermore, Zheng et al. 2019 showed that 50 nm polystyrene nanoparticles increased the level of malondialdehyde (MDA) in rat hepatocytes, being a marker of lipid peroxidation [51]. Other studies reported a significant increase in lipid peroxidation in the muscle and brain tissue of sea bass (*Dicentrarchus labrax*) after 24-h exposure to microplastics at a concentration of 0.69 mg/L [18,19,52]. In turn, Ribeiro et al. (2017) demonstrated the formation of oxidative damage in *Scrobicularia plana* following exposure to 20 µm polystyrene microparticles at the concentration of 1 mg/L. This analysis demonstrated a slight increase in the LPO level [53]. Another study showed an increase in MDA in green discus fish exposed to microplastic at concentrations of 0, 50 and 500 µg/L [54]. Despite the above-mentioned studies, other authors observed that exposure to plastic microparticles did not induce lipid peroxidation in sea shellfish (*Mytilus edulis*) [55].

As a result of lipid peroxidation, aldehydes are formed, which can act cytotoxically [56] and lead to the inhibition of the activity of some transport proteins and membrane enzymes [57]. The process of protein oxidation may cause reversible and irreversible damage depending on the source of ROS and the type of oxidants [58]. Free radicals attack the main polypeptide chain and side chains of amino acid residues as a result of aromatic hydroxylation or thioloxidation [59]. Usually, aromatic amino acids (tyrosine, tryptophan and phenylalanine) and sulfur amino acids (methionine and cysteine) are damaged [58]. Cysteine and methionine, which contain active sulfur atoms, are the most susceptible to oxidation [60]. Oxidants attack the protein backbone, thereby causing protein conformational changes and fragmenting these structures. Induced by oxidation, intermolecular bridges can change the proteolytic properties of proteins and cause their aggregation [59].

Therefore, we also analyzed the level of oxidative changes in PBMCs proteins under the influence of polystyrene nanoparticles of different sizes. On the basis of the decrease in tryptophan fluorescence, we found that the smallest nanoparticles (29 nm) caused statistically significant changes at the concentration of 0.1 µg/mL, while nanoparticles with a diameter of 44 and 72 nm only at a concentration ten times higher (1 µg/mL) (Table 2).

Research by Holloczki and Gehrke (2019) showed that 5 nm nanoplastics interacted with proteins, thereby changing their key secondary structure and leading to their denaturation. It was also observed that amino acids containing side chains, i.e., tryptophan and phenylalanine, can be adsorbed on the surface of nanoplastics. Moreover, these studies suggest that a strong NP–amino acid interaction may disrupt protein folding [61]. Other studies have looked at the effect of NP on human plasma. A strong affinity of plasma proteins with the analyzed particles was demonstrated. Formation of a protein corona measuring 100 to 600 nm was observed in plasma. This analysis also showed that the interaction of proteins with nanoparticles caused conformational changes as well as protein denaturation [62]. In the latest research, scientists analyzed the effect of 150 µg/mL PS-NPs on bovine protein serum. It was proved that bovine serum albumin was maintained on the surface of the tested particles during the initial phase of internalization, which could protect the cell membrane against damage caused by polystyrene nanoparticles. Lysosomal cytotoxicity was also observed in this study due to the degradation of the protein corona [63]. 

Accumulation of ROS, oxidized protein or lipid products in the cell impairs the functions of the cell, which in turn may lead to its apoptosis [64,65,66]. Thubagere et al. (2010) suggested that the cause of the apoptosis in cells treated with NPs may be the accumulation of H_2_O_2_ [67]. Other studies have shown that accumulation of NPs in lysosomes plays a central role in the cell death, leading to swelling of the lysosomes and the release of cathepsins into the cytosol, which ultimately propagates damage to the mitochondria with subsequent activation of apoptosis [68].

In the next step, we conducted quantitative analysis of the apoptosis of PBMCs treated with PS-NPs. Our results show that PS-NPs caused a statistically significant increase in the number of apoptotic cells, which was dependent on the concentration of tested NPs. Importantly, the earliest changes were caused by the smallest NPs, starting from a concentration of 1 µg/mL (Figure 4). The concentration-dependent apoptotic effect of PS-NPs has also been demonstrated in the studies conducted on the human epithelial carcinoma cell line (A431) [69] and RAW264.7 cells [70]. In another study, researchers showed that PS-NPs induced apoptosis of human lung epithelial cells A549. They observed that PS-NPs induced the significant up-regulation of pro-apoptotic proteins such as DR5, caspase-3, caspase-8, caspase-9 and cytochrome c, which revealed that PS-NPs triggered a TNF-α-associated apoptosis pathway [71].

At the end, we determined the percentage of viable cells after 24 incubations of PBMCs with polystyrene nanoparticles of different sizes, and we made an attempt to determine the lethal concentration (LC_50_). Our research has shown that nanoparticles with a size of 29 and 44 nm cause a statistically significant decrease in viability from the concentration of 500 µg/mL by 25% and 20%, respectively, while nanoparticles with a size of 72 nm cause a similar statistically significant decrease in viability only at the concentration of 1000 µg/mL. We also determined the LC_50_ for PBMCs under the influence of polystyrene nanoparticles of different sizes. LC_50_ for nanoparticles with a size of 29 nm was 957 µg/mL, while for nanoparticles with a size of 44 and 72 nm it was >1000 µg/mL (Figure 5). Rubio et al. (2020) also investigated changes in the viability of three selected cell-lines after PS-NP exposure. The cells THP-1, TK6 and Raji-B were exposed for 24 and 48 h to a range of the concentrations of PS-NPs, up to 200 μg/mL. The authors observed only mild cytotoxic effects of studied particles at their highest tested concentration in two types of cells: TK6 and Raji-B [47].

Summing up, we found the pro-oxidative effect of the analyzed PS-NPs in human PBMCs. The effect was dependent on the diameter of the NPs, which may affect their different ability to enter the cells. The largest effects were shown for the smallest NPs, of 29 nm. A similar effect was observed by Xu et al. (2019), who incubated the human alveolar epithelial A549 cell line with NPs of 25 and 70 nm. The authors showed that smaller PS-NPs were rapidly internalized by the cell and caused greater changes in the tested parameters [71].

Due to the occurrence of oxidative, unfavorable processes caused by polystyrene nanoparticles, their accumulation in the tissues of mammals and humans may have negative long-term consequences, and requires further detailed research.

## 4. Materials and Methods

### 4.1. Biological Material

The study was conducted on peripheral blood mononuclear cells. Peripheral human blood mononuclear cells were collected from a leucocyte–buffy coat from the blood of healthy donors. Blood was collected by employees of Blood Bank in Lodz, Poland, from volunteering donors and then was subjected to laboratory diagnostics. PBMCs were isolated from the leucocyte–buffy coat separated from blood bought from the Regional Centre of Blood Donation and Blood Treatment in Lodz, Poland. Blood was collected from 25 healthy individuals (non-smoking donors with no known illness, aged 18–30). Department of Biophysics of Environmental Pollution, University of Lodz, purchases blood for research based on the agreement of Blood Donation and Healing. Blood Bank in Lodz is accredited by Health Minister (No BA/2/2004) in the field of taking blood and separating its ingredients. The research was approved by the Bioethics Committee of the University of Lodz (Resolution No. 8/KBBN-UŁ/II/2019 (08.04.2019)).

### 4.2. Chemical Standards 

Standards of PS-NPs (diameter: 29, 44 and 72 nm) were purchased from Polysciences Europe GmbH. PBS buffer was used to dissolve compounds. Fluorescent markers, i.e., 3′-(p-hydroxyphenyl)-fluorescein (HPF) to measure the level of the hydroxyl radical, 6-carboxy-2′,7′dichlorodihydro-fluorescein diacetate (H_2_DCFDA-AM) used for ROS analysis, propidium iodide (PI) and calcein-AM for cell viability assessment and cis-parinaric acid were purchased from the company Molecular Probes (USA). FITC Annexin V Apoptosis Detection Kit (BD Pharmingen™) was purchased from BD Biosciences (USA). Lymphocyte separation medium (LSM) (1.077 g/cm) and RPMI 1640 medium were bought from Biotech. Potassium chloride (KCl), sodium chloride (NaCl), sodium hydrogen phosphate (Na_2_HPO_4_), potassium dihydrogen phosphate (KH_2_PO_4_), ammonium chloride (NH_4_Cl), sodium hydrogen carbonate (NaHCO_3_), ethylenediaminetetraacetic acid (EDTA) and other chemicals were purchased from Sigma.

### 4.3. Characterization of Polystyrene Nanoparticles

The average diameters of PS-NPs were measured by dynamic light scattering (DLS) using Zetasizer Nano-ZS (Malvern Instruments, Malvern, UK). Samples were measured at 25 °C in phosphate-buffered saline (PBS), pH 7.4, in plastic cuvettes. The analysis was made using the Malvern Instruments software. The refraction factor for PBS was assumed to be 1.50. Detection wavelength was 633 nm, and detection angle was 90°.

### 4.4. Obtaining Leucocytes

The buffy coat was centrifuged (3000 rpm, 10 min, at 20 °C) to remove plasma. Then, the lymphocyte layer was harvested. PBMCs were isolated using LSM, a mixture of Ficoll^®^ and sodium diatrizoate (Hypaque) (density 1.077 g/mL) by centrifugation at 600× *g* for 30 min at 20 °C. After centrifugation, PBMCs were harvested and resuspended in 3 mL of red blood cell lysis buffer and incubated for 5 min at 20°C, then PBS was added, followed by centrifugation at 200× *g* for 15 min at 20 °C. The supernatant was collected and cells deposited on the bottom of the tube were washed with RPMI medium containing L-glutamine and 10% fetal bovine serum and subjected to another centrifugation at 200× *g* for 15 min. The final density of cells used for the study was 1 × 10^6^ cells/mL [72].

### 4.5. Determination of Reactive Oxygen Species Levels

Changes in the level of reactive oxygen species formed were assessed based on the analysis of the oxidation of 5′carboxy-2′,7′-dichlorodihydrofluorescein diacetate. This marker is nonpolar and therefore it can enter a cell. While penetrating the cell membrane, it is cleaved by membrane esterase to a nonfluorescent compound dichlorodihydrofluorescein, which under the influence of ROS is oxidized to 2′,7′-dichlorofluoerescein (DCF), emitting green fluorescence. After 24 h of incubation of PBMCs with a suspension of polystyrene nanoparticles, the samples were centrifuged (3000 rpm, 10 min, at 4 °C) and suspended in PBS and the marker was added to the final concentration of 2.5 µM. The mixture was incubated for 15 min in the dark at 37 °C. ROS level analysis was performed with the flow cytometer (LSR^®^ II from Becton-Dickinson, San Jose, CA, USA) at an excitation wavelength of 490 nm and an emission of 530 nm for a total number of 10,000 counts [73].

### 4.6. Determination of Hydroxyl Radical Level

The principle of the analysis of the level of hydroxyl radical is the oxidation of the 3′-(p-hydroxyphenyl)-fluorescein probe as a result of the interaction with highly reactive oxygen species (mainly hydroxyl radicals). The reaction produces fluorescein that emits green fluorescence at the wavelength of 515 nm [74]. After 24 h of incubation of PBMCs with a suspension of polystyrene nanoparticles, samples were centrifuged (3000 rpm, 4 °C) and were suspended in PBS, and the marker was added to the final concentration of 2 µM. Prepared samples were then incubated for 15 min at 37 °C in the dark. The flow cytometer (LSR^®^ II from Becton-Dickinson, San Jose, CA, USA) was used for a total number of 10,000 events.

### 4.7. Lipid Peroxidation 

Lipid peroxidation was analyzed by measuring the fluorescence of cis-parinaric acid at an excitation of 320 nm and emission of 432 nm. In the first step, cis-parinaric acid was added to the suspension of lymphocytes, to obtain the final concentration of 5 µM in sample, and incubated for 1 h at 37 °C in the dark, which allowed incorporation of the acid into the cell membrane [75]. After 24 h of incubation of PBMCs with a suspension of polystyrene nanoparticles, samples were centrifuged (3000 rpm, 4 °C) and were suspended in PBS, and the marker was added to the final concentration of 2 µM. Samples were centrifuged (600× *g*, 10 min, at 4 °C) to remove an excess of cis-parinaric acid. Remaining cells were suspended in RPMI medium and subjected to another 24-h incubation with analyzed compounds. The above analysis was performed using a 96-well microplate reader (Cary Eclipse Fluorescence Spectrophotometer–Varian, Australia).

### 4.8. Protein Oxidation

Protein oxidation was assessed by measuring tryptophan fluorescence at an excitation of 295 nm and emission of 335 nm. Fluorescent properties of proteins result from the presence of aromatic amino acids in their structure, e.g., tryptophan [76]. A decrease in fluorescence of the analyzed samples resulted from oxidative damage of tryptophan, as well as damage of proteins in PBMCs membrane. After 24 h of incubation of PBMCs with a suspension of polystyrene nanoparticles, samples were centrifuged (3000 rpm, 4 °C) and were suspended in PBS, and the marker was added to the final concentration of 2 µM. Samples were centrifuged (600× *g*, 10 min, at 4 °C). Supernatant was discarded and lymphocytes were suspended in RPMI medium with L-glutamine. Protein damage analysis was performed in 96-well plates using the microplate reader (Cary Eclipse Fluorescence Spectrophotometer–Varian, Australia).

### 4.9. Detection of Apoptosis

The samples were incubated with a suspension of polystyrene nanoparticles in final concentrations ranging from 0.1 to 500 μg/mL for 24 h at 37 °C in total darkness. The test was carried out in accordance with the instructions of the manufacturer. The cells were stained with PI and fluorescein conjugated with Annexin V in Annexin-binding buffer for 15 min at room temperature in total darkness. The samples were analyzed by flow cytometry (LSR^®^ II from Becton-Dickinson, San Jose, CA, USA) with excitation at 488 nm to visualize fluorescein and PI fluorescence at maxima of 525 and 617 nm, respectively [72].

### 4.10. Analysis of Cells Viability of PBMC

Cell viability analysis was performed with calcein-AM and propidium iodide. Calcein-AM is hydrolyzed to a hydrophilic, highly fluorescent calcein, which is retained inside living cells [77]. In living cells, calcein ester emits strong green fluorescent light at an excitation of 490 nm and an emission of 515 nm, which demonstrates its fluorescent properties that calcein itself does not have. PI, on the other hand, is a marker of dead cells with two positive charges. This dye enters dead cells and binds to DNA, emitting red fluorescent light, which can be excited by a wavelength of 535 nm and an emission of 617 nm. After 24 h of incubation of PBMCs with a suspension of polystyrene nanoparticles, samples were centrifuged (3000 rpm, 4 °C) and were suspended in PBS, and the marker was added to the final concentration of 2 µM. Samples were centrifuged (100× *g*, 10 min, 4 °C) and then the marker mix (PI and calcein-AM) was added. Finally, the samples were incubated in the dark for 15 min, transferred to ice, and their fluorescence was measured. The total number of counts per sample was 10,000. Cell viability analysis using the above stains was performed on the flow cytometer (LSR^®^ II from Becton-Dickinson, San Jose, CA, USA).

### 4.11. Statistical Analysis

The assays were performed on blood from 5 donors (5 experiments were conducted), whereas for each donor, the experimental point was a mean value of at least 3 replications. The results are shown as mean ± SD. Multiple comparisons among the group mean differences were analyzed by one-way analysis of variance (ANOVA) followed by Tukey’s post-hoc test. Tukey test was used as a post-hoc test. When the p value was lower than 0.05, the differences were considered statistically significant (*). Statistical analysis was conducted using STATISTICA software ver.13 (StatSoft Inc., Tulsa, OK, USA).

## 5. Conclusions

This study for the first time illustrates the action of polystyrene nanoparticles on human PBMCs.

Polystyrene nanoparticles increased ROS levels, including hydroxyl radical, as well as induced the oxidation of lipids and proteins.

The pro-oxidative effects of PS-NPs resulted in an increase in apoptosis and decrease in PBMC viability.

Observed changes in PBMCs incubated with PS-NPs depended on their size. A comparison of the actions of PS-NPs showed that the smallest nanoparticles (29 nm) exhibited the strongest oxidative potential in PBMCs

Changes in the parameters studied occurred at very low PS-NPs concentrations (0.01 µg/mL), which may potentially be found in the human body as a result of environmental exposure.

## Figures and Tables

**Figure 1 ijms-22-04406-f001:**
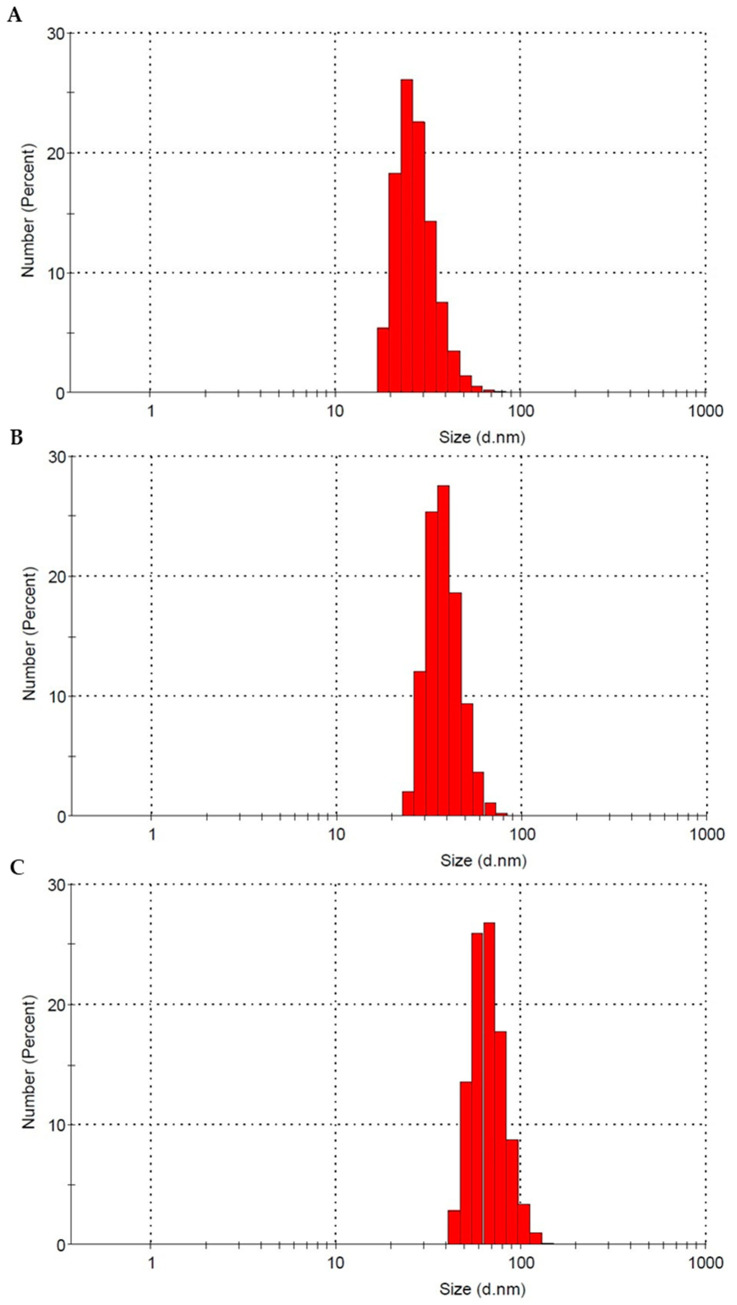
Dynamic light scattering size distribution of the different PS-NPs in phosphate-buffered saline, pH 7.4. Graphs, (**A**)—29 nm, (**B**)—44 nm and (**C**)—72 nm, were prepared by means of the Zetasizer Nano-ZS software.

**Figure 2 ijms-22-04406-f002:**
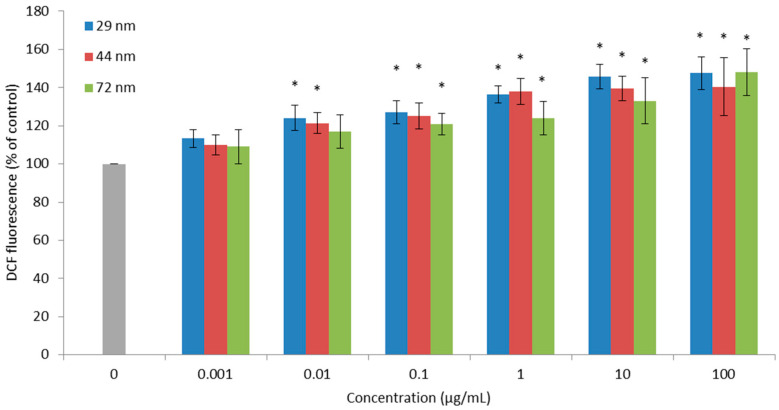
ROS level in PBMCs incubated with PS-NPs of diameter 29, 44 and 72 nm in the concentration range of 0.001 to 100 µg/mL for 24 h (n = 5). * *p* < 0.05 indicates statistically significant difference from control; one-way ANOVA and a posteriori Tukey test.

**Figure 3 ijms-22-04406-f003:**
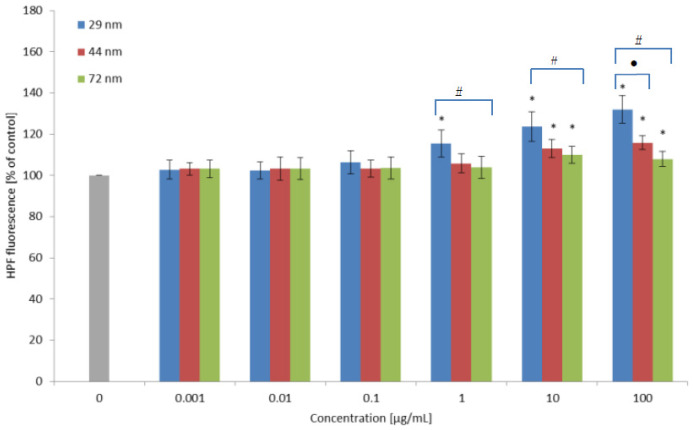
Changes in hydroxyl radical levels of cells incubated with PS-NPs of diameter 29, 44 and 72 nm in the concentration range of 0.001 to 100 µg/mL for 24 h (n = 5). * *p* < 0.05 indicates statistically significant difference from control; (•) statistically significant difference between nanoparticles of size 29 versus 44 nm; (#) statistically significant difference between nanoparticles of size 29 versus 72 nm. One-way ANOVA and a posteriori Tukey test.

**Figure 4 ijms-22-04406-f004:**
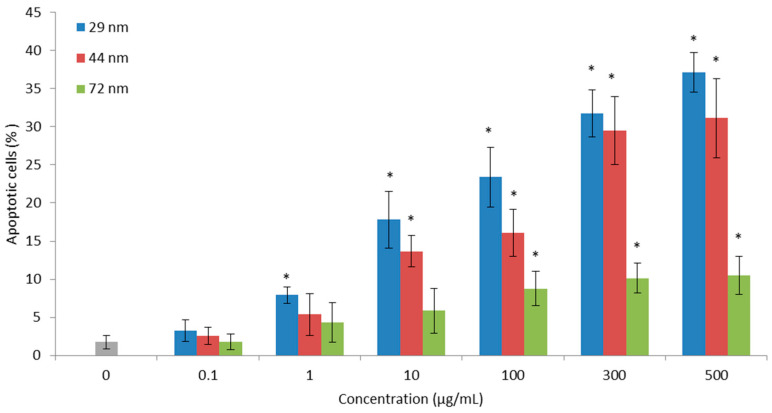
Percentage of apoptotic PBMCs incubated for 24 h with PS-NPs of diameter 29, 44 and 72 nm in the concentration range from 0.01 to 500 µg/mL (n = 5). * *p* < 0.05 indicates statistically significant difference from control; one-way ANOVA and a posteriori Tukey test.

**Figure 5 ijms-22-04406-f005:**
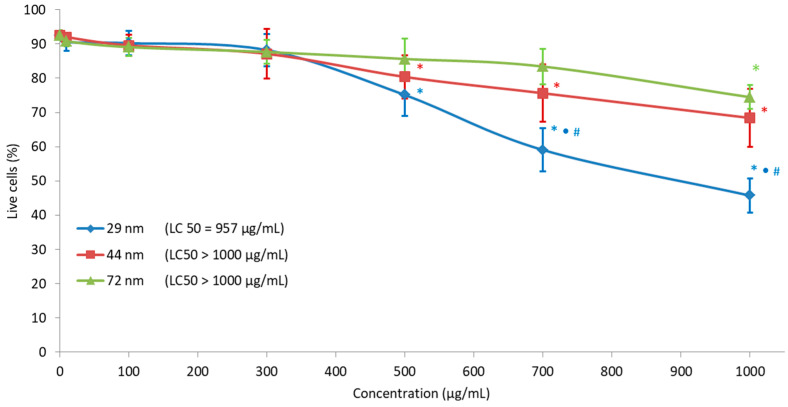
Percentage of cell viability incubated with PS-NPs with a diameter of 29, 44 and 72 nm in the concentration range of 10 to 1000 µg/mL for 24 h (n = 5). * *p* < 0.05 indicates statistically significant difference from control; (•) statistically significant difference between nanoparticles of the size 29 versus 44 nm; (#) statistically significant difference between nanoparticles of the size 29 versus 72 nm. One-way ANOVA and a posteriori Tukey test.

**Table 1 ijms-22-04406-t001:** Characterization of PS-NPs size by DLS in phosphate-buffered saline (PBS), pH 7.4.

Diameter in Dicated by Manufacturer	29 nm	44 nm	72 nm
Mean Diameter ± SD	27.96 ± 8.03	38.61 ± 8.61	68.45 ± 15.19

**Table 2 ijms-22-04406-t002:** The level of lipid peroxidation (fluorescence of cis-parinaric acid) and protein oxidation (tryptofan fluorescence) in PBMCs incubated with PS-NPs of diameter 29, 44 and 72 nm in the concentration range of 0.001 to 100 µg/mL for 24 h (n = 5).

Concentration (µg/mL)	Fluorescence of Cis-Parinaric Acid (% of Control)			Tryptofan Fluorescence (% of Control)		
	29 nm	44 nm	72 nm	29 nm	44 nm	72 nm
0	100	100	100	100	100	100
0.001	96.83 ± 5.09	97.86 ± 5.86	98.30 ± 4.99	95.65 ± 5.28	96.28 ± 6.36	98.38 ± 7.34
0.01	94.80 ± 6.50	95.81 ± 6.27	95.13 ± 6.91	94.39 ± 6.71	95.97 ± 7.01	97.46 ± 5.71
0.1	83.30 ± 5.60 *	94.00 ± 6.83	95.31 ± 6.99	86.19 ± 5.82 *	94.66 ± 6.71	98.56 ± 7.51
1	79.04 ± 4.77 *	91.03 ± 5.89 *	89.46 ± 4.41 *	82.88 ± 7.01 *	89.29 ± 7.25 *	95.06 ± 6.57 *
10	76.21 ± 4.63 *	83.32 ± 5.45 *	85.56 ± 5.93 *	83.00 ± 5.53 *	87.92 ± 4.45 *	83.75 ± 5.42 *
100	74.23 ± 4.08 *	77.29 ± 7.45 *	74.22 ± 5.60 *	83.31 ± 4.47 *	84.88 ± 5.98 *	78.86 ± 4.81 *

* *p* < 0.05 indicates statistically significant difference from control; one-way ANOVA and a posteriori Tukey test.

## Data Availability

Not applicable.

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
