# Peer review of "Oxidative Properties of Polystyrene Nanoparticles with Different Diameters in Human Peripheral Blood Mononuclear Cells (In Vitro Study)"

_ijms, 2021, doi:10.3390/ijms22094406_

Round 1
Reviewer 1 Report
The manuscript analysed the effect of polystyrene nanoparticles (29, 44 and 72 nm diameter) on human peripheral blood mononuclear cells. The in vitro study showed the changes on ROS levels, lipid and protein oxidation, and cell viability due to the presence of nanoparticles. In vitro studies are necessary to elucidate the cellular response to potential toxic compounds. In addition, the use of in vitro analyses instead of in vivo ones is a key point to reduce the use of animals for explorative cytotoxic studies.
The innovative issue of the manuscript is convincing. The study is well designed, carried out and described, however the following major and minor comments could improve the manuscript:
- Figure legends specify the incubation time of nanoparticles with cells (24 h), these data should be added to the Material and Methods section for each technique performed.
- Cell viability was performed after 24 h of incubation. However, it is known that low ROS levels could undergo apoptosis pathways. The method used to detect cell viability does not allow to detect cells that have already started apoptosis. In addition to 24 h experiment, I recommend to analyse the cell viability after 48 h or to analyse an apoptotic marker after 24 h.
- The discussion includes several references about in vivo studies, other in vitro analyses of the effect of microparticles and nanoparticles should be added.
- The authors described that nanoparticles can be internalized by different mechanisms according the size. Passive diffusion of plasma membranes has been described for small nanoparticles. Could the results observed be directly related to the ease of internalization?
- Some acronyms like HPF or H2DCFA are defined more than once, please check.
Author Response
Dear Reviewer 1,
I am very grateful to the Reviewer for a thorough reading of my paper and numerous remarks and suggestions regarding the mistakes I had committed. These remarks were very much to the point and allowed me to take a critical look at the paper. I have corrected the paper in the line with the Reviewer remarks, and I hope that it has become more clear now.
Responds to the reviewer’s comments:
Figure legends specify the incubation time of nanoparticles with cells (24 h), these data should be added to the Material and Methods section for each technique performed.
Response: We added information about the incubation time of PBMCs with PS-NPs to each method performed.
2. Cell viability was performed after 24 h of incubation. However, it is known that low ROS levels could undergo apoptosis pathways. The method used to detect cell viability does not allow to detect cells that have already started apoptosis. In addition to 24 h experiment, I recommend to analyse the cell viability after 48 h or to analyse an apoptotic marker after 24 h.
Response: As suggested by the Reviewer, we added the results showing the percentage of apoptotic PBMCs after incubation with PS-NPs for 24 h.
3. The discussion includes several references about in vivo studies, other in vitro analyses of the effect of microparticles and nanoparticles should be added.
Response: As suggested by the Reviewer, in the discussion, we added references concerning the effect of PS-NPs on cells (in in vitro system).
4. The authors described that nanoparticles can be internalized by different mechanisms according the size. Passive diffusion of plasma membranes has been described for small nanoparticles. Could the results observed be directly related to the ease of internalization?
Response: We think so, because small nanoparticles should penetrate the cell more easily.
5. Some acronyms like HPF or H2DCFA are defined more than once, please check.
Response: Repeated acronyms have been removed, and now they are cited only once in the paper.

Reviewer 2 Report
The manuscript entitled “Oxidative properties of polystyrene nanoparticles with different diameters in human peripheral blood mononuclear cells (in vitro study).” is interesting and well organized. Please consider reviewing based on the suggestions below, in order to improve the article.
Results
-I understand that the authors purchased the polystyrene nanoparticles with different dimensions, but it is recommended to add a dimensional analysis to the manuscript (DLS, TEM or SEM).
Materials and Methods
-On sections 4.4., 4.5, 4.6, 4.7, 4.8 authors said that: “After incubation of PBMCs with polystyrene nanoparticles…..”. Please explain in which form was used the nanoparticles when were put in contact with PBMCs (dry state or as suspension?).
-It is recommended to add some conclusions in order to increase the quality of the manuscript.
Author Response
Dear Reviewer 2,
I am very grateful to the Reviewer for thorough reading of my paper and numerous remarks and suggestions regarding the mistakes I had committed. These remarks were very much to the point and allowed me to take a critical look at the paper. I have corrected the paper in the line with the Reviewer remarks, and I hope that it has become more clear now.
Responds to the reviewer’s comments:
1. I understand that the authors purchased the polystyrene nanoparticles with different dimensions, but it is recommended to add a dimensional analysis to the manuscript (DLS, TEM or SEM).
Response: As suggested, we conducted analysis by means of Dynamic Light Scattering (DLS).
The obtained results were added into the manuscript (‘Results’ section).
2. On sections 4.4., 4.5, 4.6, 4.7, 4.8 authors said that: “After incubation of PBMCs with polystyrene nanoparticles…..”. Please explain in which form was used the nanoparticles when were put in contact with PBMCs (dry state or as suspension?).
Response: In the study, nanoparticles were used in the form of a suspension. As suggested, we added this information in sections 4.4 - 4.9.
3. It is recommended to add some conclusions in order to increase the quality of the manuscript.
Response: The conclusions were added in a separate chapter "Conclusions ''.

Round 2
Reviewer 1 Report
The authors have addressed all the comments and the manuscript has been improved. I recommend the manuscript for publication.